# Biofilm-coated microbeads and the mouse ear skin: An innovative model for analysing anti-biofilm immune response *in vivo*

Léo Sauvat[1,2], Aizat Iman Abdul Hamid[1], Christelle Blavignac[3], Jérôme Josse[4], Olivier Lesens[1,2], Pascale Gueirard[1] *

1 Laboratoire Microorganismes: Génome et Environnement, Université Clermont Auvergne, UMR CNRS 6023, Clermont-Ferrand, France, 2 Infectious and Tropical Diseases Department, CRIOA, CRMVT, CHU Clermont-Ferrand, Clermont-Ferrand, France, 3 Centre Imagerie Cellulaire Santé, Université Clermont Auvergne, Clermont-Ferrand, France, 4 CIRI–Centre International de Recherche en Infectiologie, Inserm, U1111, CNRS, UMR5308, École Normale Supérieure de Lyon, Université Lyon, Université Claude Bernard Lyon 1, Lyon, France

* pascale.gueirard@uca.fr

**Data Availability Statement:** All relevant data are within the manuscript and its Supporting Information files.

**Funding:** PG Pack research Ambition 2017-IMMUNOFILM-Staph project (Auvergne Rhones

## Abstract

Owing to its ability to form biofilms, *Staphylococcus aureus* is responsible for an increasing number of infections on implantable medical devices. The aim of this study was to develop a mouse model using microbeads coated with *S. aureus* biofilm to simulate such infections and to analyse the dynamics of anti-biofilm inflammatory responses by intravital imaging. Scanning electron microscopy and flow cytometry were used *in vitro* to study the ability of an mCherry fluorescent strain of *S. aureus* to coat silica microbeads. Biofilm-coated microbeads were then inoculated intradermally into the ear tissue of LysM-EGFP transgenic mice (EGFP fluorescent immune cells). General and specific real-time inflammatory responses were studied in ear tissue by confocal microscopy at early (4-6h) and late time points (after 24h) after injection. The displacement properties of immune cells were analysed. The responses were compared with those obtained in control mice injected with only microbeads. *In vitro*, our protocol was capable of generating reproducible inocula of biofilm-coated microbeads verified by labelling matrix components, observing biofilm ultrastructure and confirmed *in vivo* and *in situ* with a matrix specific fluorescent probe. *In vivo*, a major inflammatory response was observed in the mouse ear pinna at both time points. Real-time observations of cell recruitment at injection sites showed that immune cells had difficulty in accessing biofilm bacteria and highlighted areas of direct interaction. The average speed of cells was lower in infected mice compared to control mice and in tissue areas where direct contact between immune cells and bacteria was observed, the average cell velocity and linearity were decreased in comparison to cells in areas where no bacteria were visible. This model provides an innovative way to analyse specific immune responses against biofilm infections on medical devices. It paves the way for live evaluation of the effectiveness of immunomodulatory therapies combined with antibiotics.

Alpes Region) https://aaprecherche.
auvergnerhonealpes.fr/ The funders had no role in
study design, data collection and analysis, decision
to publish, or preparation of the manuscript.

**Competing interests:** The authors have declared
that no competing interests exist.

## Introduction

The implantation of invasive medical devices occurs routinely in almost all fields of medicine. However, these medical procedures considerably increase the risk of microbial infections for patients. In hospitals, the prevalence of nosocomial infections is therefore high (about 5%), with 32% of clinical cases being caused by foreign body-related infections (FBRIs) [1]. *Staphylococcus aureus* (*S. aureus*) is the second most incriminated microorganism in nosocomial infections and the most incriminated pathogen in surgical site infections (SSIs) after prosthetic joint implantations [2]. These infections require heavy, costly and sometimes functionally deleterious surgery, associated with long-term antibiotic treatments [3]. A prosthetic joint infection, for example, will multiply by three the initial cost of surgery [4].

Like many other bacterial species, *S. aureus* is able to adhere to different types of biotic/abiotic surfaces, promoting the formation of microbial communities called biofilms. This property is one of the most important features of its pathogenicity [5]. Biofilms represent one of the two radically different lifestyles that bacteria can adopt, as opposed to the isolated and free-floating form in liquid medium called planktonic bacteria. Biofilms are the most common form of bacteria and are responsible for 80% of human chronic bacterial infections in developed countries [6]. Thus, understanding the pathophysiology of *S. aureus* FBRIs and developing preventive strategies for SSIs are currently active fields of research [7].

One key element of *S. aureus* pathophysiology is the ability of biofilms to circumvent host immune attacks [8]. During *S. aureus* cutaneous infections, tissue-resident mast cells recruit neutrophils and monocytes/macrophages from the bloodstream [9] along with monocytic myeloid-derived suppressor cells (M-MDSCs) in interaction with natural killer (NK) cells to promote the first level of inflammatory responses [10]. *S. aureus* is able to decrease these innate immune responses by acting on complement activation and phagocyte recruitment, and by secreting immune-evasion proteins [8]. The planktonic to biofilm developmental transition also represents one of the primary immune-evasion mechanisms that allow neutralization of neutrophil effector functions [11]. Finally, the complexity of biofilm structure contributes to the increased resistance of biofilm bacteria to host immune defences and to appropriate antibiotic treatments, giving rise to the concept of "biofilm-related disease" [12, 13].

The few studies performed *in vivo* to analyse the inflammatory response to *S. aureus* biofilm bacteria are still in the development phase and the dynamic aspects of the responses are almost unexplored [14]. A better understanding of the different biofilm immune evasion mechanisms is therefore required to develop suitable therapeutic strategies [15].

A mouse ear skin model was previously designed to analyse the dynamics of innate immune responses against *S. aureus* biofilms. In this model, biofilms were micro-injected into the mouse ear pinna, a cutaneous tissue widely used to analyse dynamic interactions between microorganisms and resident or recruited myeloid cells by intravital imaging [16]. This mouse model evidenced inflammatory responses specific to biofilms. The aim of our work was to improve the model by introducing a support made from a comparable material to that used for orthopedic prostheses, so as to mimic biofilm infections on medical devices in humans. We chose silica microbeads as miniaturized inert supports because they have a ceramic-like structure and are compatible with the intradermal micro-injection of biofilms into the ear pinna of LysM-EGFP transgenic mice. In a first step, we optimized the preparation of biofilm-coated microbeads and then analysed biofilm properties on the surface of microbeads. We further compared the dynamics of inflammatory responses against biofilm-coated microbeads or against a mixture of uncoated microbeads and planktonic bacteria injected into the mouse ear pinna.

## Materials and methods

### Materials

**Mice and ethical statement.**   C57BL/6 WT mice (6–8 week-old males and females) were purchased from Charles River Laboratories. LysM-EGFP transgenic mice (6–8 week-old males and females) were obtained from the bacteria-cell interactions unit of the Pasteur Institute (Paris, France). Mice were bred in the animal care facility at Clermont Auvergne University (Clermont-Ferrand, France). All experiments were approved by the local Ethics Committee on Animal Experimentation (Auvergne C2E2A, Clermont-Ferrand, France, agreement number: 1725) and were carried out in accordance with the applicable guidelines and regulations. All mice were provided an appropriate environment including shelter in cages, a comfortable resting area, sufficient space (not more than 5 animals per cage) and ready access to fresh water and food to maintain full health and vigour. Animal welfare was observed on a daily basis to ensure optimal conditions and treatment which avoid suffering. Littermates destined to be inoculated were housed in separate cages with access to the same facilities previously stated. The anaesthetic used during experiments was chosen in order to promote deep anaesthesia. During and after imaging sessions, mice were kept warm in order to prevent any risks related to hypothermia. Euthanasia by cervical dislocation on the anesthetized animal was performed at the end of the infection period.

**mCherry and GFP-tagged strain construction.**   The *S. aureus* SH1000 mCherry-tagged strain (named mCherry-SH1000) was constructed after insertion of the pCtuf-mCherry plasmid [17] by electroporation into the SH1000 strain isolated from a corneal ulcer [18], as described previously [19]. The mCherry-SH1000 strain was then selected onto Luria-Bertani (LB) agar containing chloramphenicol (10 μg/mL). Clones were grown overnight with shaking in Trypticase Soja (TS) culture medium containing chloramphenicol (10 μg/mL) and stored at −80°C in the same medium with 15% glycerol. Fluorescence was detected in bacterial suspensions by fluorescence microscopy. The *S. aureus* SH1000 GFP-tagged fluorescent strain (named GFP-SH1000) was constructed in the same way with the pCN47-GFP plasmid [20] but using erythromycin (10 μg/mL) in place of chloramphenicol.

**Microbeads.**   As an abiotic support for the preparation of biofilms (Monodisperse Silica Microspheres, Cospheric), we chose silica microspheres (microbeads) with the following characteristics: 2.0g/cc, d50 = 4.3μm, CV = 2.7%, <1% doubles. The absence of local or systemic toxicity of the silica microbeads was ascertained after intravenous injection in mice [21]. The stock sample consisted of a mixture of 50 mg of microbead powder from the commercial sample in 2 mL of PBS.

### Biofilm preparation and analysis

**Inoculum preparation.**   GFP and mCherry fluorescent SH1000 strains were used in all the experiments performed. Planktonic bacterial cultures were prepared from an aliquot of frozen bacteria in Trypticase Soy (TS) culture medium supplemented with chloramphenicol or erythromycin for mCherry-SH1000 and GFP-SH1000 strains, respectively. The bacterial culture was placed overnight at 37°C with agitation under aerobic conditions.

Planktonic mCherry-SH1000 were prepared from overnight cultures. After homogenization, the bacterial concentration (Colony Forming Unit/mL or CFU/mL) of overnight cultures was estimated by measuring the Optical Density at a wavelength of 600 nm ($OD_{600nm}$) and multiplying it by the known titre of the strain ($1.35 \times 10^8$ CFU/$OD_{600nm}$ units). The corresponding volume of overnight culture containing $5 \times 10^6$ CFUs was withdrawn and centrifuged at

3000 g for 5 minutes. The pellet was resuspended in 3.8 μL of Phosphate-Buffered Saline (PBS) containing $10^5$ microbeads.

Biofilm inocula (mCherry and GFP-SH1000) were prepared after adjusting an overnight culture to $OD_{600nm} = 1 \pm 0.05$ in TS culture medium. The suspension was then diluted 100x and placed in at least two flat-bottomed-wells in a 24-well cell culture plate (1 mL per well). Microbeads were added to each well ($4x10^6$), which were then placed at 37°C in a humidity chamber for 24 hours at 20 rpm of horizontal agitation. The biofilm formed on the surface of the microbeads was then steam-washed for 40 minutes as described previously [22]. Biofilms were recovered by flushing and scraping the bottom of the well in 200 μL of PBS. This suspension was then transferred to a second steam-washed biofilm well using the same biofilm recovery technique. The resulting suspension was washed twice by centrifuging at 100 g and then resuspending the pellet in 1 mL of PBS at room temperature. After the final wash, the inoculum was resuspended in 50 μL of PBS, of which 3.8 μL (corresponding to $10^5$ microbeads) were used as the inoculum.

Consequently, at this stage of preparation, 3 kinds of inocula were microinjected: "planktonic bacteria and microbeads", "biofilm-coated microbeads" and the control group consisting of $10^5$ microbeads suspended in 3,8 μL of PBS.

In order to quantify planktonic or biofilm inocula by titration, the samples were first diluted 10x in PBS. The diluted biofilm was vortexed for 30 seconds, sonicated for 10 minutes and then vortexed again for a further 30 seconds (Fisher Scientific, 80W, 37kHz). Inocula titrations were determined by serial dilutions and plating on LB agar medium containing chloramphenicol or erythromycin. CFUs were counted after 24 hours of incubation at 37°C.

Control inocula consisting of microbeads resuspended in PBS were obtained by diluting the stock sample to a concentration of $2.6x10^4$ microbeads/μL of PBS, verified by KOVA® cell counts, of which 3.8 μL were used as the inoculum.

**Observation of bacterial inocula by fluorescence microscopy, flow cytometry and Scanning Electron Microscopy (SEM).** The coating efficiency of biofilms on microbeads was first studied by fluorescence microscopy (LEICA MM AF microscope, objective X63). Biofilms formed on microbeads by the mCherry-SH1000 strain were observed before and after passing through the 34G injection needle.

For flow cytometry analysis, samples of each type of inocula were studied with a FACSAria Fusion SORP (BD BioSciences) equipped with FACSDIVA 8 software (BD Sciences). A 100 μm nozzle and a 1.5 neutral density filter were used. Excitation sources were 488 and 561 nm lasers. Emission of mCherry fluorescence was collected with a set of 600nm Long Pass and 610/20 nm Band Pass filters. All signals were collected and analysed on logarithmic scales. The threshold was set at 200 on the Forward Scatter (FSC) parameter. The gating strategy first excluded cell debris. Beads were then gated on Forward/ Side Scatter (FSC/SSC) parameters and bacteria were gated as a "non-bead" population. Data were acquired for one minute at flow rate 4 (30μL/minute).

For SEM analysis, mCherry-SH1000 biofilms were prepared during the process described above with 3 different samples. Two kinds of inoculum of "biofilm-coated microbeads", were prepared as described above. The first kind of sample was analysed before inoculum passing through the 34-gauge (34G) needle used for micro-injection into the mouse ear tissue and the second after passing through the needle. The comparison between these 2 samples was necessary in order to confirm that the morphological characteristics of the inoculum are maintained after micro-injection. The last kind of sample was studied after sonication in order to confirm that biofilm aggregates were absent from the inoculum. Microdroplets of this sample were deposited on SEM Pore filters (DTM9305, Jeol) either with a 34G needle fitted to a NanoFil syringe (World Precision Instruments) (after micro-injection) [23] or a pipette (before micro-

injection) and passively diffused (slowly/gently) through it. After absorption on the filters, the bacteria were fixed for 12 hours at 4˚C in 0.2 mol/L sodium cacodylate buffer, pH 7.4, that contained 1.6% glutaraldehyde. They were then washed 30 minutes in sodium cacodylate buffer (0.2 mol/L, pH 7.4) and post-fixed 1 hour with 1% osmium tetroxide in the same buffer. After rinsing for 20 minutes in distilled water, dehydration by graded ethanol was performed from 25˚ to 100˚ in progressive steps of 10 minutes to finish in hexamethyldisilazane (HMDS) for 10 minutes. Samples were then sputter-coated with gold-palladium (JFC-1300, JEOL, Japan). Observations were made with a JSM-6060LV scanning electron microscope (Jeol, Japan) at 5kV in high-vacuum mode.

**Matrix labelling with WGA-Alexa 488 and the CDy11 fluorescent probe.**   Biofilms of mCherry-SH1000 were prepared according to the protocol described. Just after the steam wash, 200 μL of Wheat Germ Agglutinin (WGA)-Alexa 488 (5 μg/mL in PBS) were added to the wells and incubated for 10 minutes at 37˚C. The wells were then gently rinsed with water and scraped according to the usual inoculum preparation protocol. After diluting and spreading of 10 μL of the solution on a glass slide, the samples were analysed on the Leica MMAF Imaging System Microscope at X20.

Labelling with the CDy11 fluorescent probe [16] was carried out on GFP-SH1000 biofilms. Samples were prepared as previously described. A 10 mM stock solution of the fluorescent probe was prepared in dimethyl sulfoxide (DMSO). The solution was then diluted in PBS to prepare a 100 μM solution. *In vitro* labelling was carried out by adding 10 μL of the diluted probe to biofilm samples, which were gently vortexed and incubated for 45 minutes in the dark at room temperature. Marked samples were then micro-injected into the ear pinna of mice. After 30 minutes, the inner side of the ear was flattened on a glass slide as previously described [24] and observed on the ZEISS Spinning Disk Cell Observer (SD) (Carl Zeiss Microscopy, Germany) confocal microscope. Video acquisition was carried out with two different lasers to observe GFP and CDy11 fluorescence (excitation at 488 and 590 nm, emission at 509 and 612 nm, with exposure times set at 100 and 300 ms, respectively). Images were acquired with the 10X objective lens. Multiple fields of observation were imaged to observe the entire injection site. *In vivo* labelling was carried out by the same method except for the 45-minute incubation period.

## Intravital imaging by confocal microscopy

**Time-lapse video and mosaic acquisition.**   Mice were anaesthetized by intraperitoneal injection of a ketamine and xylazine mixture. Planktonic or biofilm inocula or PBS were injected intradermally as previously described, into the ear tissue of LysM-Enhanced Green Fluorescent Protein or LysM-EGFP transgenic mice [23]. Three to five hours after injection of the inocula, the animals were again anaesthetized. Mosaic and video time-lapse acquisition was carried out on the ZEISS LSM 800 (LSM 800) (Carl Zeiss Microscopy, Germany) and SD confocal microscopes, respectively, as previously described [25].

**Analysis of *in vivo* confocal imaging.**   Images acquired on the LSM 800 were stitched together with ZEN software to reconstruct an entire image of the ear tissue at early and late time points. The images shown represent the Z-projected maximal intensity signal of a reconstituted image of the ear tissue for the EGFP channel. Images were then analysed, as previously described [25].

Images and videos acquired on the SD with the 10X objective lens were first stitched using ZEN software. Images shown correspond to the Z-projected maximum intensity signal for each channel. Time-lapse videos at 20X and 10X were analysed with Imaris software as previously described, to analyse two different parameters (average speed and straightness) of immune cell dynamics [25].

**Statistical analysis.** Data generated were analysed with Prism 5 software (GraphPad Software, Inc.) and a non-parametric Mann-Whitney two-tailed statistical test. $p \leq 0.05$ was considered statistically significant (symbols: ****$p \leq 0.0001$; ***$p \leq 0.001$; **$p \leq 0.01$; *$\leq 0.05$; ns = non-significant).

## Results

### Preparation and characterization of *Staphylococcus aureus* biofilm-coated microbeads

The first set of experiments enabled us to obtain calibrated inocula of mCherry-SH1000 biofilms coated on an abiotic surface. The selected inert surfaces were 4.3 $\mu m^2$ microbeads, compatible with the micro-injection of very small volumes of bacterial inocula into the mouse ear pinna, as previously described [25]. A number of crucial steps were optimized to obtain standardized preparations of biofilm-coated silica microbeads. First, as microbeads moved around in their wells, horizontal agitation was slowed down to allow the planktonic bacteria to adhere and form persistent biofilms on the surface of microbeads. Biofilm-covered microbeads were gently steamed to eliminate free-floating planktonic bacteria and also to prevent biofilms covering the microbeads from becoming detached. Different sets of uncoated (Fig 1A and 1B) or coated microbeads were imaged by confocal microscopy and by SEM to analyse biofilm coverage. After washing and homogenization, the suspension of potentially biofilm-coated microbeads was analysed at the ultrastructural level by SEM. It mainly contained microbeads trapped inside biofilm aggregates (Fig 1C and S1A Fig), and also some small clusters of detached or planktonic bacteria. Adherent viable fluorescent bacteria were also clearly visible at the surface of the microbeads by confocal microscopy (Fig 2A). At high magnification by SEM, the extracellular matrix was clearly visible at the microbead surface (Fig 1D and S1B–S1D Fig). In particular, glucidic matrix components were detected at the surface of the microbeads after labelling with WGA and observation by confocal microscopy (Fig 2A).

In parallel to the qualitative analysis carried out by SEM, we also wanted to quantitatively characterize our biofilm inoculum. To do this, a flow cytometry protocol was designed to quantify the proportion of biofilm-coated microbeads in the suspension. The number of mCherry fluorescent bacteria associated with microbeads was compared in three different samples: uncoated microbeads, a freshly prepared mix of uncoated microbeads and planktonic bacteria, and biofilm-coated microbeads. The acquisition of a great number of events ($3.7 \times 10^6$) showed a high percentage (82.5%) of biofilm–coated microbeads in our inocula compared to that of control samples (Fig 2B–2D).

Thus, our biofilm inoculum, presenting the characteristics previously described, was hereafter referred to as "biofilm-coated microbeads". Fig 1E and 1F show that the morphological characteristics of the inoculum are maintained after passing through the 34-gauge (34G) needle used for micro-injection into the mouse ear tissue. We also set up a sonication protocol to titrate our inoculum. When used on biofilm-coated microbeads, bacterial aggregates were absent from the surface of microbeads, reflecting biofilm dispersion, with only a few adherent bacteria attached to microbeads (Fig 1G and 1H), a result that therefore validates our inoculum titration protocol.

In a second set of experiments, the biofilm-coated microbead inoculum was characterized *in vivo*. A 3.8 $\mu l$ calibrated inoculum containing $10^5$ biofilm-coated microbeads was microinjected into the ear pinna of LysM-EGFP transgenic mice. In parallel, control groups of mice received $10^5$ uncoated microbeads mixed with planktonic bacteria just before micro-injection, hereafter designated "Planktonic bacteria & microbead" inoculum. Titration of both types of inocula revealed comparable titres, equal to $2.04 \times 10^7 \pm 1.49 \times 10^7$ and $1.97 \times 10^7 \pm 3.07 \times 10^7$,

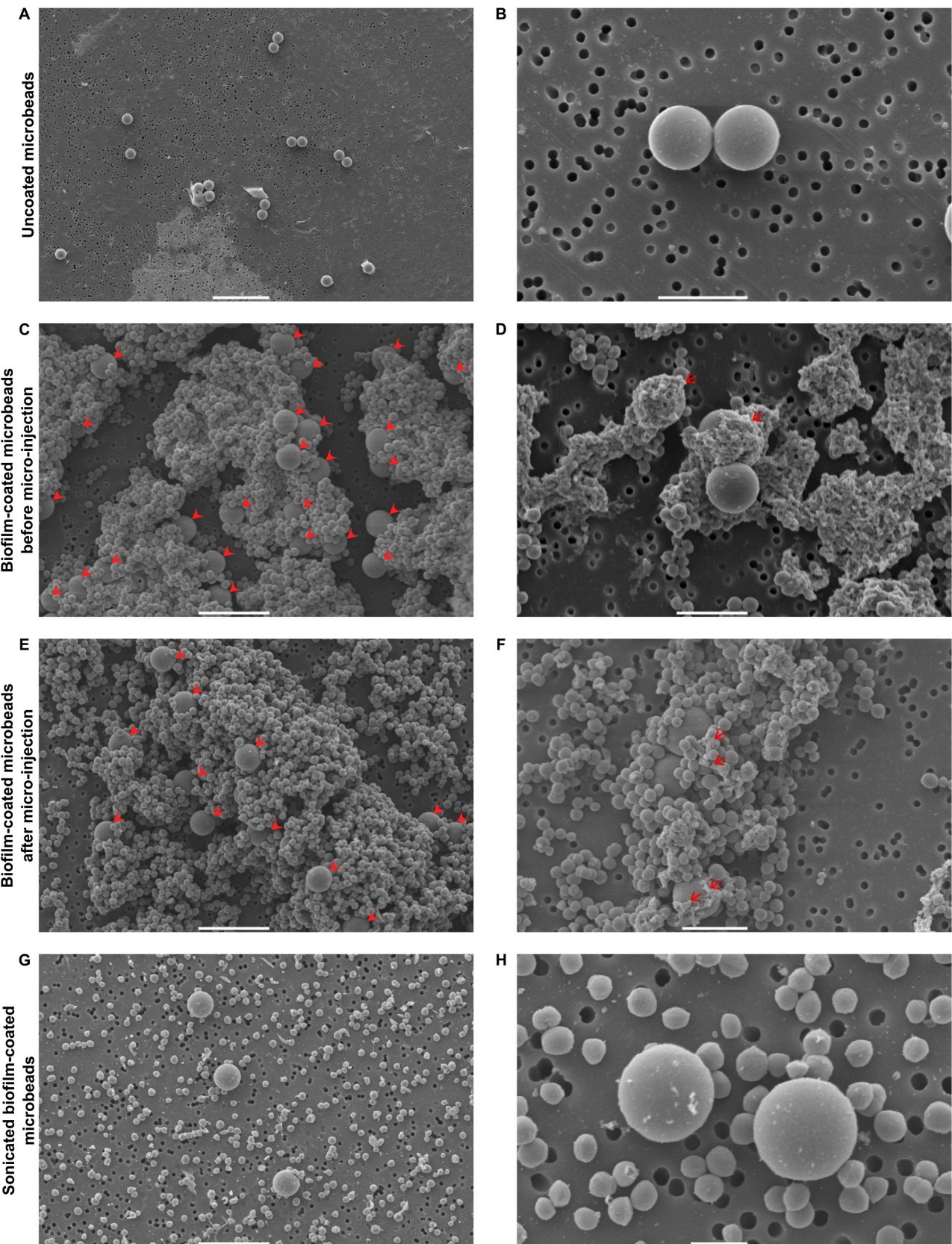

**Fig 1. SEM micrographs of uncoated and biofilm-coated microbeads before and after microinjection, and after sonication.** SEM micrographs of uncoated microbeads at X800 (**A**) and X5000 (**B**) magnification, biofilm-coated microbeads before passing through the 34G needle used for microinjections at X2000 (**C**) and X4000 (**D**) magnification, biofilm-coated microbeads after passing through the 34G needle used for microinjections at X2000 (**E**) and X3700 (**F**) magnification, and sonicated biofilm-coated microbeads at X2000 (**G**) and X8000 (**H**) magnification. Filled red arrowheads in **C** and **E** indicate microbeads and red arrows in **C** and **E** indicate biofilm extracellular matrix. Scale bar: 20 µm (**A**), 10 µm (**C, E, G**), 5 µm (**B, D, F**), 2 µm (**H**).

respectively, for biofilm-coated microbeads and planktonic bacteria & microbeads. Fluorescent bacteria coated on microbeads were visualized at the injection site by confocal microscopy as previously described [25]. Matrix components (amyloid fibrils) were also detected *in vivo* on biofilm-coated microbeads pre-incubated/pre-labelled with the fluorescent probe CDy11 *in vitro* (Fig 3B). Co-localization was sometimes observed between the green signal (bacteria) and the red signal (matrix components). In the next step, amyloid fibrils were labelled *in situ* by co-injecting biofilm-coated microbeads and the CDy11 fluorescent probe into the ear pinna. As illustrated in Fig 3B (*in vitro* labelling) and 3c (*in vivo* labelling), comparable images were observed at the injection site, in terms both of intensity of the fluorescent red signal and of fluorescence co-localization (as compared to the control in Fig 3A). This novel finding means that, using our model, it is possible for the first time to follow biofilm development *in vivo* by injecting the CDy11 fluorescent probe into the cutaneous tissue and *in situ* by labelling matrix components inside biofilm-coated microbeads.

## Characterization of inflammatory responses after the microinjection of *Staphylococcus aureus* biofilm-coated microbeads in the mouse ear pinna

The mouse ear skin model previously developed to analyse immune responses against *S. aureus* biofilms was redesigned to make it better adapted to the context of biofilm infections on medical devices. After the addition of microbeads to the biofilm cultures, imaging protocols were applied to the new model to compare inflammatory responses at the tissue and cellular levels [25]. LysM-EGFP transgenic mice were inoculated intradermally with $10^5$ biofilm-coated microbeads, with a mix of $10^5$ uncoated microbeads and a theoretical preparation of $10^7$ CFU of planktonic bacteria, or with PBS containing $10^5$ uncoated microbeads as a control.

The inflammatory response was explored in the entire tissue at early (6h) and late (26h) time points by analysing the EGFP signal corresponding to the recruitment of innate immune cells at the injection site, as previously described [25]. At both time points, EGFP+ cells were recruited in infected mice (Fig 4A–4C), with a highly significant increase between early and late time points, and no significant difference between the "Planktonic bacteria & microbead" group and the "Biofilm-coated microbead" group at late time points. At early time-points, the EGFP signal wads statistically significant only for the biofilm group (Fig 4D).

The inflammatory response was further analysed at the cellular level at early time points (4 to 6h post-infection) using the intravital imaging approach and the previously devised confocal acquisition protocol. The aim of this second set of experiments was to study the impact of the presence of microbeads on the dynamics of innate immune responses against *S. aureus* biofilms by comparing the results with our previous data [25]. In control mice inoculated with PBS, low recruitment was observed at the injection sites, owing to microinjection trauma (S1 Movie). An influx of EGFP+ phagocytic cells was observed in the two infected groups, "Planktonic bacteria & microbeads" and "Biofilm-coated microbeads" (Fig 5A and 5B, white circles, S2 and S3 Movies), and the phenotype of our previous observations without microbeads was reproduced. The recruited cells covered most of the injection area when planktonic bacteria were inoculated, with multiple contact points between cells and bacteria (Fig 5A, white arrowheads, Fig 5C and 5D) while behaving differently for the biofilm-coated microbead inoculum.

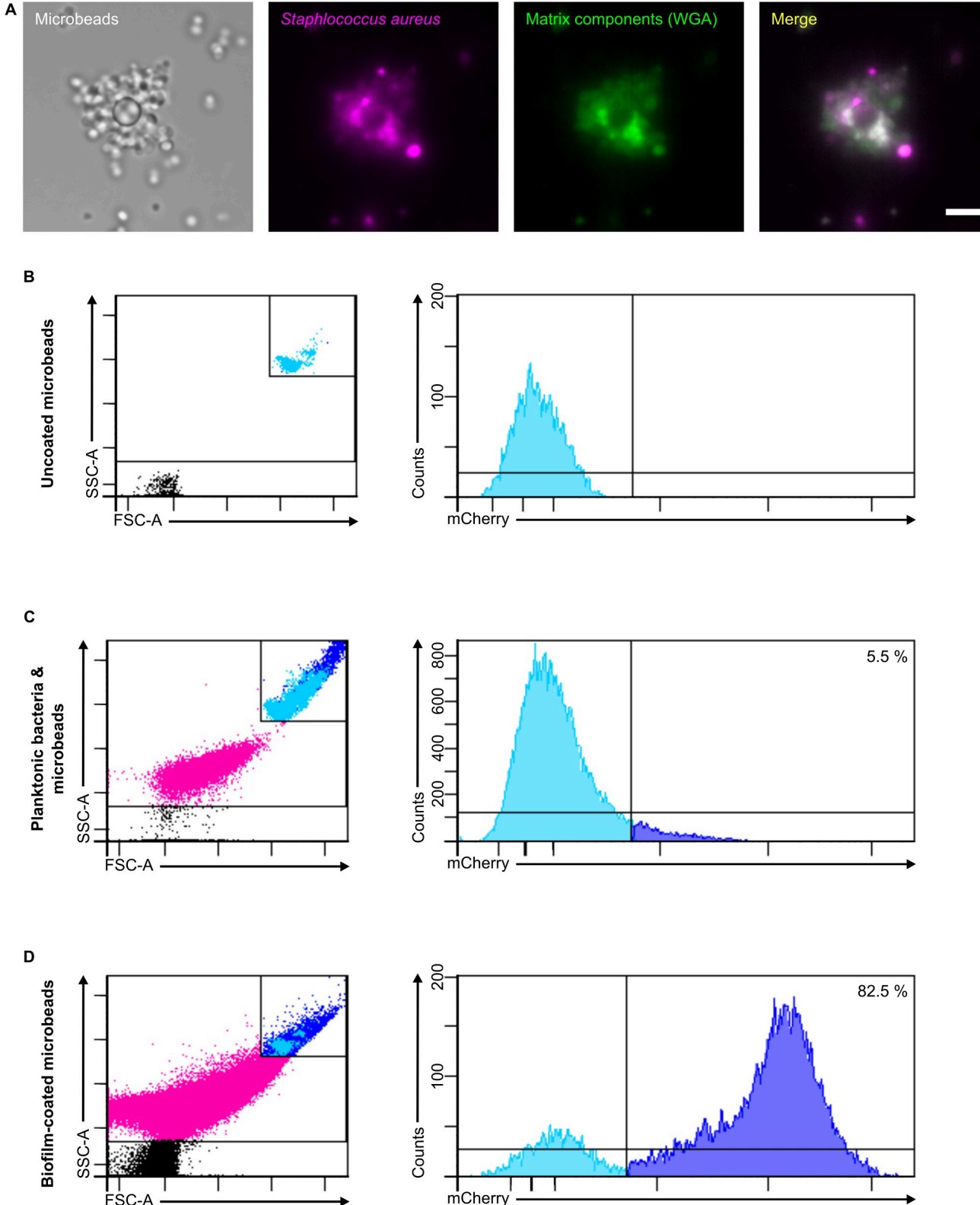

**Fig 2. Confocal images of a biofilm-coated microbead and representative flow cytometry plots verifying coating of microbeads.** Confocal images **(A)** of biofilm-coated microbeads, after passing through the micro-injection needle, treated with Alexa Fluor 488 WGA at X63 magnification. Images show microbeads and maximum intensity projections of mCherry (magenta) and Alexa Fluor 488 WGA (green) fluorescence. The merged image shows maximum intensity projections for both fluorescence channels. Scale bar: 5 μm. Representative FACS plots showing the gates used to determine uncoated

microbeads **(B)**, planktonic bacteria and microbeads **(C)** and biofilm-coated microbead **(D)** populations. The subsequent counts for each population are also presented according to their fluorescence intensities.

The cells clearly remained at the periphery of the inoculum, and the contact points were less numerous (Fig 5B, white arrowheads). Thus, use of biofilm-coated microbeads and of the intravital imaging approach identified a specific phenotype for the dynamics of innate immune responses against biofilms.

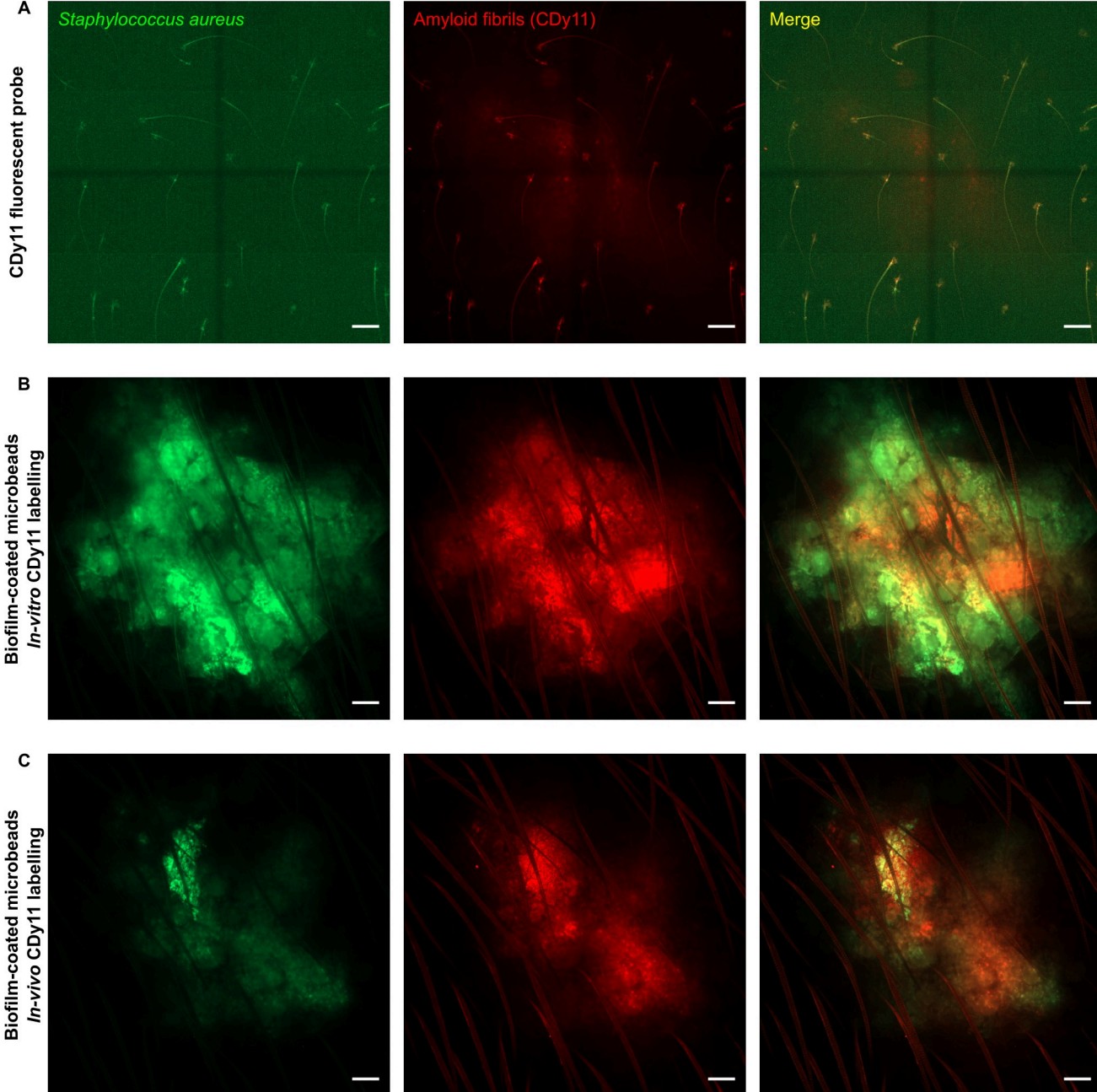

**Fig 3. Intravital confocal imaging of CDy11 labelled biofilm-coated microbeads.** Intravital confocal imaging after microinjection of only the CDy11 fluorescent probe **(A)**, of biofilm-coated microbeads after *in vitro* CDy11 labelling **(B)** or of biofilm-coated microbeads after *in vivo* CDy11 labelling **(C)** in the ear pinna of WT C57BL/6 mice. **(A to C)** Images show maximum intensity projections of GFP (green) and CDy11 (red) fluorescence. The merged image shows maximum intensity projections for both fluorescence channels. Scale bar: 100 μm.

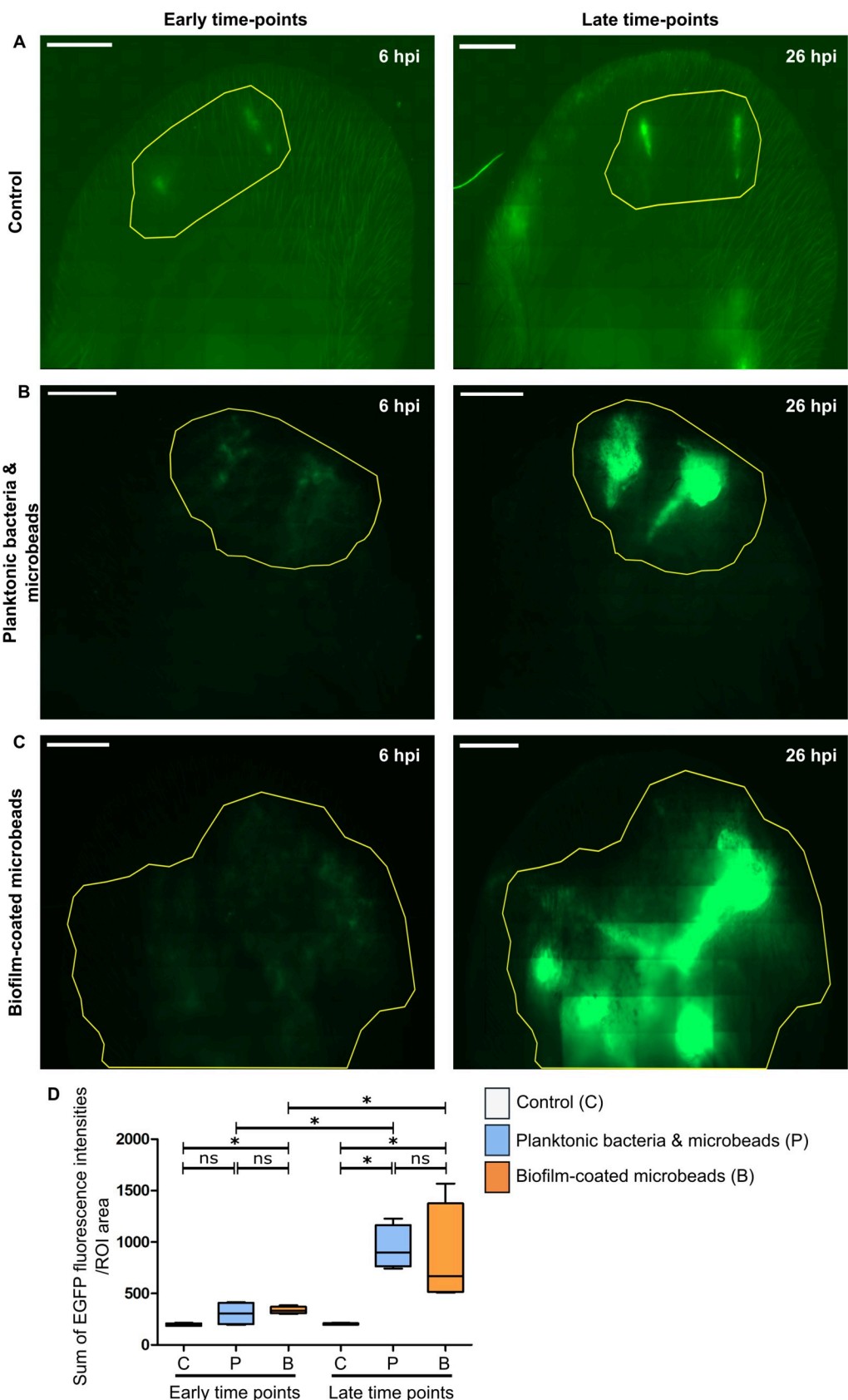

**Fig 4. Global inflammatory response after microinjection of different inocula into the transgenic mice ear.**
Reconstituted confocal images of LyM-EGFP transgenic mice ear pinna tissue after microinjection of either PBS (control) **(A)**, planktonic bacteria and microbeads **(B)** or biofilm-coated microbeads **(C)** at early (6 hours post-infection or 6 hpi) and late time points (after 26 hpi). Images show maximum intensity projections of EGFP (green) fluorescence that correspond to phagocytic cells (neutrophils and macrophages). The yellow line indicates the Region of Interest (ROI) where the "Sum of EGFP fluorescence intensities" was measured. One representative experiment is shown for each group of mice from four independent experiments. Scale bar: 2 mm. **(D)** Ratio of the sum of EGFP fluorescence intensities to ROI area. Data are expressed as median and interquartile ranges for four mice per group. $p \leq 0.05$ was considered statistically significant (symbols: ****$p \leq 0.0001$; ***$p \leq 0.001$; **$p \leq 0.01$; *$\leq 0.05$; ns = non-significant).

## Analysis of the dynamic properties of immune cell recruitment after microinjection of *Staphylococcus aureus* biofilm-coated microbeads into the mouse ear pinna

In the last set of experiments, the previously designed tracking protocol was used to analyse the motility properties of recruited EGFP+ cells [25]. The average speed and straightness of the trajectory of phagocytic cells were analysed for the three groups of mice at early time points (4 to 6h post-infection). Average speed was significantly lower for the two bacterial forms than in control mice (Fig 6A), indicating that phagocytic cells were arrested at the injection site to interact with bacteria, irrespective of the bacterial form. However, the straightness of their trajectories was significantly decreased only for the biofilm-coated microbead group (Fig 6B). A separate analysis of the motility properties of cells interacting (bacteria contact) or not (no bacteria contact) with bacteria showed that for both bacterial forms the cells significantly slowed down when in contact with bacteria. Interestingly, cells not in contact with bacteria had significantly lower speed values in the biofilm group than in the planktonic group (Fig 6C). When there was no interaction with bacteria, the trajectory straightness of the recruited cells was significantly greater for both bacterial forms (Fig 6D). Taken together, these findings enabled us to identify the specific cell dynamics of the inflammatory response against the biofilm-coated microbead inoculum and show that our model is a powerful tool to analyse the way in which biofilms can circumvent host immune attacks.

## Discussion

In this work, we devised a new ear skin model using microbeads coated by *S. aureus* biofilm to mimic biofilm infections on medical devices in humans. The model allowed us to analyse the dynamics of the inflammatory responses against *S. aureus* biofilms. Rodent laboratory models previously designed to analyse these responses used devices such as silicone implants and catheters that were pre-colonized or not [14]. To mimic *S. aureus* SSI of prosthetic joints in mice, implants [26] or steel pins colonized before insertion in the tibial region have also been used [27]. We chose microbeads as a support because their micrometric size is compatible with the mouse ear skin model. Previous studies described the specific adhesion of *S. aureus* biofilms on glass beads [28] or of *P. aeruginosa* biofilms on ceramic beads [29]. *In vitro*, 5mm diameter glass beads were also used to evaluate the efficacy of different antiseptic solutions against *P. aeruginosa* biofilm [30]. We adapted this protocol to standardize the different steps of biofilm preparation according to the microbead support (biofilm culture, analysis of biofilm formation, use of sonication). To have neutral material such as the ceramides used in orthopedic prostheses, we selected 4.3 μm diameter silica dioxide microbeads. This choice allowed us to inoculate very small volumes of biofilm-coated microbeads into the ear tissue of mice [25]. Other groups of mice received the same volume of a mix containing an equivalent CFU number of planktonic bacteria and of uncoated microbeads, as in the biofilm-coated microbead inoculum. The major difference between the two inocula was the presence of matrix

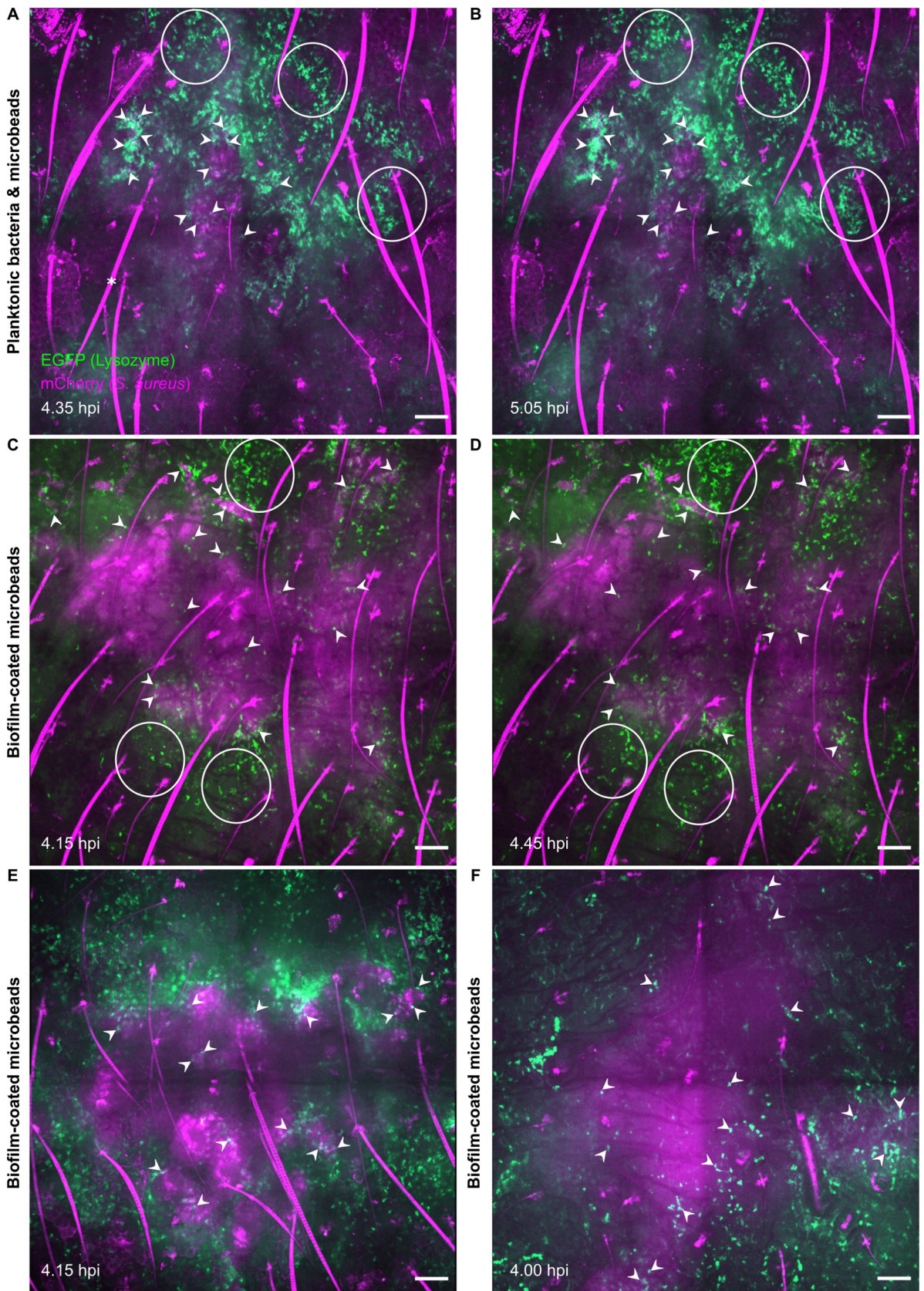

**Fig 5. Live confocal imaging of innate immune responses after microinjection of the different inocula.** Live confocal imaging after microinjection of planktonic bacteria and microbeads **(A, B)** in the ear pinna of LysM-EGFP transgenic mice at early time points (4–6 hpi). Innate immune cell recruitment towards injection sites was observed between 4.35 **(A)** to 5.05 hpi **(B)**. Live confocal imaging after microinjection of biofilm-coated microbeads **(C, D, E, F)** in the ear pinna of LysM-EGFP transgenic mice at early time points (4–6 hpi). Innate immune cell recruitment towards injection sites was observed between 4.15 **(C)** to 4.45 hpi **(D)**. Images show maximum intensity projections of EGFP (green) and mCherry (magenta) fluorescence. A progressive recruitment of EGFP+ innate immune cells was observed (white empty circles) at the injection sites with cell-bacteria contact areas (filled white arrowheads). *: autofluorescent hair (also in magenta). Filled white arrowheads indicate cell-bacteria contact. Live confocal imaging of two additional experiments at early time points after microinjection of biofilm-coated microbeads, at 4.15 hpi **(E)** and 4.00 hpi **(F)**. Scale bar: 100 μm.

components detectable at the surface of biofilm-coated microbeads by both SEM and the CDy11 probe. The latter is a particularly valuable tool to detect matrix components such as amyloid fibrils *in vivo*. The model we developed enabled us to compare immune responses in the skin specific to *S. aureus* planktonic form and to biofilm coated on an abiotic support. Using the CDY11 probe, we were also able to detect biofilms at the surface of microbeads *in situ* and in the cutaneous tissue, and to follow its development *in vivo*.

After inoculation, the inflammatory response to *S. aureus* was analysed at the tissue and cellular levels over time. At the tissue level, our new model showed that both bacterial forms induced a measurable inflammatory response at cutaneous injection sites in LysM-EGFP transgenic mice, as compared to control mice. EGFP+ phagocytic cells, namely neutrophils and monocyte/macrophages, were recruited and detected after 6h, and the inflammatory

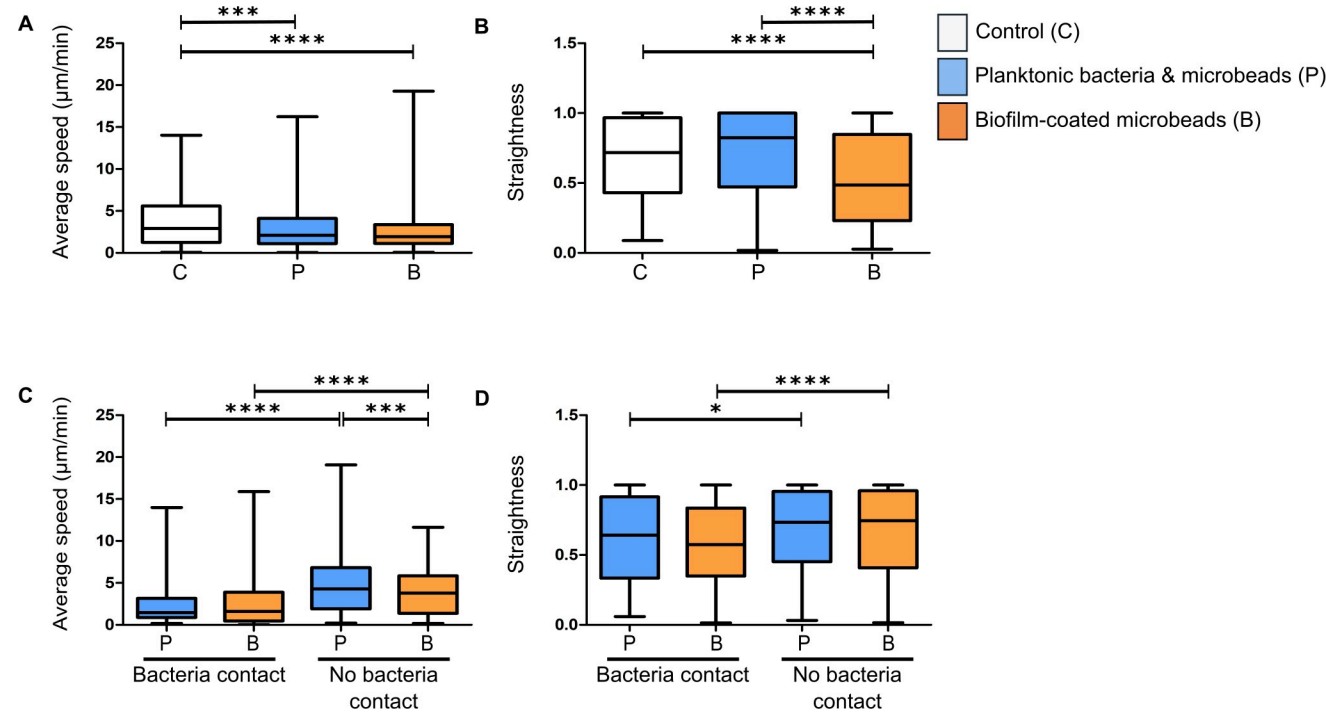

**Fig 6. Analysis of average speed and straightness of innate immune cells in the mouse ear pinna.** Average speed **(A and C)** and straightness **(B and D)** of EGFP+ cells recruited to injection sites at early time points after microinjection of PBS (control), planktonic bacteria and microbeads, or biofilm-coated microbeads. Data are expressed as median and interquartile ranges pooled from three different mice in three independent experiments for each group. Average speed **(A)** and straightness **(B)** of all cells (in contact with visible bacteria or not) in infected and control mice. Number of cells (N) analysed for each group at early time points: Control: N = 190 cells; Planktonic form: N = 721 cells; Biofilm form: N = 771. Average speed **(C)** and straightness **(D)** of cells either in contact (bacteria contact) or not (no bacteria contact) with planktonic or biofilm bacteria at early time points. Number of cells (N) analysed at early time points that were in contact or not in contact with bacteria, respectively: Planktonic form: N = 238 and 386 cells; Biofilm form: N = 433 and 276 cells. p≤0.05 was considered statistically significant (symbols: ****p≤0.0001; ***p≤0.001; **p≤0.01; *≤0.05; ns = non-significant).

response was marked and comparable for both inocula after 26h. These results are consistent with those of previous *in vivo* studies analysing qualitatively and quantitatively the inflammatory responses against *S. aureus* biofilm in rodents, and sometimes the associated cytokine and chemokine profile [14, 31–35]. In most models, an implanted infected support was used (K-wire, catheter, pin) and large numbers of neutrophils and monocytes/macrophages were observed to migrate.

At the cellular level, using the intravital imaging approach, we confirmed our previous observations regarding the dynamics of recruitment of EGFP+ cells. A specific phenotype was clearly identified for the biofilm-coated microbead group of mice. Recruited cells had difficulty in accessing biofilm bacteria and the contact areas between phagocytes and bacteria were less numerous than in the planktonic group. These results illustrate two major characteristics of biofilm physiology: the notion that biofilms represent a physical barrier for phagocytes [31] and that consequently phagocytosis is ineffective [36].

Our new ear skin model finally enabled us to quantify the motility parameters (speed, straightness) of EGFP+ recruited phagocytes at the injection site through a maximum thickness of around 150 μm. The analysis was performed at the tissue injection site taking into account technical parameters such as fluorescence intensity of cells, cell location and cell number [25]. For each cell, the trajectory calculated by the Imaris software was checked manually point by point. Continuous movements of each cell were also followed throughout the acquisition time to ensure that there were no automatic assignment errors by the software. The exact trajectory of immune cells was difficult to analyse when the cells were too numerous, as previously reported [37]. Analysis of the entire population of cells at the injection site (cells interacting with bacteria or not) showed that both inocula modified cell motility parameters by decreasing cell speed, as compared to the control group. However, decreased straightness of trajectory was observed only in the biofilm-coated microbead group, which indicates that the biofilm inoculum had a greater effect on cell motility than planktonic bacteria, even when the latter were co-injected with uncoated beads, because it modified both cell speed and straightness. When cells interacted with bacteria, both inocula decreased cell speed and straightness, so that engulfment and phagocytosis of bacteria (adherent or not to microbeads) occurred in all cases. This observation is consistent with previous results obtained with uncoated biofilms [25]. In the absence of visible cell-bacteria contact, the decrease in cell speed was greater in the biofilm-coated microbead group, suggestive of a biofilm effect at a distance from the recruited phagocytes. Recent reviews have reported both a direct effect of biofilm by physical interactions and an indirect effect (small molecules) on recruited immune cells [8]. The staphylococcal complement inhibitor is one example of a soluble factor that can diffuse at the early stage of biofilm formation and act as an immunomodulator [38].

In conclusion, we devised a new ear skin model of *S. aureus* biofilm infection using microbeads as a support for bacterial development. Using the intravital imaging approach, we provide evidence that the biofilm-coated microbead inoculum induces a different qualitative and quantitative inflammatory response, respectively, at the tissue and cellular levels, than the planktonic inoculum. This novel *in vivo* model of *S. aureus* biofilm infection on a material support in combination with an analysis of the dynamics of innate immune responses opens up various development perspectives. In particular, it paves the way to a better understanding of the immunobiology of biofilms and will make it possible to test in real time the efficacy of new anti- *S. aureus* biofilm therapies [39, 40].

## Supporting information

**S1 Fig. SEM micrographs of biofilm-coated microbeads.** SEM micrographs of biofilm-coated microbeads at X2000 (**A**), X4000 (**B**), X7000 (**C**) and X8000 (**D**) magnification. Filled

red arrowheads in **(A)** indicate microbeads and red arrows in **(B), (C)** and **(D)** indicate biofilm extracellular matrix. Scale bar: 10 μm **(A)**, 5 μm **(B)**, 2 μm **(C)**, 2 μm **(D)**.
(TIF)

**S1 Movie. Imaging of immune cell migration after microbead injection.** *In vivo* confocal time-lapse imaging of immune cell migration in LysM-EGFP transgenic mice ear tissue injected with microbeads in PBS from 5.30 hpi to 5.55 hpi. Maximum projections of time-lapse images. Z-stacks collected 77.73 seconds apart. Scale bar: 100 μm.
(MP4)

**S2 Movie. Imaging of immune cell migration after planktonic and microbeads injection.** *In vivo* confocal time-lapse imaging of immune cell migration in LysM-EGFP transgenic mice ear tissue injected with planktonic bacteria and microbeads from 4.35 hpi to 5.05 hpi. Maximum projections of time-lapse images. Z-stacks collected 77.11 seconds apart. Scale bar: 100 μm.
(MP4)

**S3 Movie. Imaging of immune cell migration after biofilm-coated microbeads injection.** *In vivo* confocal time-lapse imaging of immune cell migration in LysM-EGFP transgenic mice ear tissue injected with biofilm-coated microbeads from 4.15 hpi to 4.44 hpi. Maximum projections of time-lapse images. Z-stacks collected 51.92 seconds apart. Scale bar: 100 μm.
(MP4)

**S1 Table. Raw data of inocula titrations.** Table presenting raw data used for the preparation of calibrated *Staphylococcus aureus* biofilm-coated microbeads and planktonic inocula.
(XLSX)

**S2 Table. Raw data used for Fig 4D.** Table presenting raw data used to measure the ratio of the sum of EGFP fluorescence intensities to ROI areas.
(XLSX)

**S3 Table. Raw data used for Fig 6A.** Table presenting the average speed of all cells in infected and control mice at early time points. Raw data extracted from Imaris software.
(XLSX)

**S4 Table. Raw data used for Fig 6B.** Table presenting the straightness of all cells in infected and control mice at early time points. Raw data extracted from Imaris software.
(XLSX)

**S5 Table. Raw data used for Fig 6C.** Table presenting the average speed of cells in contact with bacteria or not in infected mice at early time points. Raw data extracted from Imaris software.
(XLSX)

**S6 Table. Raw data used for Fig 6D.** Table presenting the straightness of cells in contact with bacteria or not in infected mice at early time points. Raw data extracted from Imaris software.
(XLSX)

## Acknowledgments

We wish to thank Alexander R. Horswill (Department of Immunology and Microbiology, University of Colorado, Denver, USA) for pAH9 plasmid, Ivo Boneca (Bacteria-Cell Interactions Unit, Pasteur Institute, Paris, France) for the LysM-EGFP transgenic mouse line, Michael Givskov and Youg-Tae Chang for the CDy11b probe, Caroline Vachias and Pierre Pouchin

(Confocal Microscopy Facility CLIC, University Clermont Auvergne) for their help with setting up image acquisition settings and image analysis and processing, Hermine Billard for help with FACS analysis (UCA Partner CYSTEM platform), Frédéric Laurent, Alan Diot and Andréa Cara for helpful discussions, and Karim Alloui for care of the animal housing facility.

## Author Contributions

**Conceptualization:** Léo Sauvat, Aizat Iman Abdul Hamid, Jérôme Josse, Olivier Lesens, Pascale Gueirard.

**Data curation:** Léo Sauvat, Aizat Iman Abdul Hamid.

**Formal analysis:** Léo Sauvat, Pascale Gueirard.

**Funding acquisition:** Pascale Gueirard.

**Investigation:** Léo Sauvat, Christelle Blavignac, Olivier Lesens, Pascale Gueirard.

**Methodology:** Léo Sauvat, Aizat Iman Abdul Hamid, Jérôme Josse, Olivier Lesens, Pascale Gueirard.

**Project administration:** Pascale Gueirard.

**Resources:** Léo Sauvat, Aizat Iman Abdul Hamid, Olivier Lesens, Pascale Gueirard.

**Supervision:** Léo Sauvat, Aizat Iman Abdul Hamid, Pascale Gueirard.

**Validation:** Léo Sauvat, Aizat Iman Abdul Hamid, Jérôme Josse, Pascale Gueirard.

**Visualization:** Léo Sauvat, Aizat Iman Abdul Hamid, Jérôme Josse, Olivier Lesens, Pascale Gueirard.

**Writing – original draft:** Léo Sauvat, Aizat Iman Abdul Hamid, Pascale Gueirard.

**Writing – review & editing:** Léo Sauvat, Aizat Iman Abdul Hamid, Jérôme Josse, Olivier Lesens, Pascale Gueirard.

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
