## [Decision Letter · Decision Letter 0]

5 Oct 2020

PONE-D-20-26002

Biofilm-coated microbeads and the mouse ear skin: an innovative model for analysing anti-biofilm immune response

PLOS ONE

Dear Dr. Gueirard,

Thank you for submitting your manuscript to PLOS ONE. After careful consideration, we feel that it has merit but does not fully meet PLOS ONE’s publication criteria as it currently stands. Therefore, we invite you to submit a revised version of the manuscript that addresses the points raised during the review process.

We look forward to receiving your revised manuscript.

Kind regards,

Amitava Mukherjee, ME, Ph.D.

Academic Editor

PLOS ONE

Journal Requirements:

2. Please amend either the title on the online submission form (via Edit Submission) or the title in the manuscript so that they are identical.

Reviewers' comments:

Reviewer's Responses to Questions

**Comments to the Author**

1. Is the manuscript technically sound, and do the data support the conclusions?

Reviewer #1: Partly

2. Has the statistical analysis been performed appropriately and rigorously? 

Reviewer #1: No

3. Have the authors made all data underlying the findings in their manuscript fully available?

Reviewer #1: Yes

4. Is the manuscript presented in an intelligible fashion and written in standard English?

Reviewer #1: Yes

5. Review Comments to the Author

Reviewer #1: The approach of using established biofilms as initial inocula and assessing the immune response will be of significant interest to the research community. The paper is well-written. Some clarifying points are provided below that I believe will help readers follow better and derive clearer information.

--Line 64: Confirm context of the 80%. I believe it’s in developing countries, not in overall infections.

--Line 138: What is 4.106? Is it 4 x 106, or 4.16? Just clarifying.

--Lines 137 – 145: What’s the rationale for disrupting the biofilms? If they grew on the surface, why not inoculate in their state of formation, rather than collection, spinning, and resuspending? By disrupting, the basal dormant layer it could make the shift to planktonic-type metabolism and not fully reflect a biofilm state.

--Line 156: Were these biofilms imaged following the same resuspension, spinning, etc. process?

--Line 166: Same question; all processed the same prior to imaging?

--Line 186: Did the vortex process disrupt the biofilm? Do you think it was still in the same state as when growth completed?

--Line 228: How does steaming prevent biofilm detachment? This may be in the literature; I’m not familiar with it.

--Lines 253 – 257: This paragraph is confusing. Did you want the biofilms on or off the beads? As questioned above, it’s not fully clear in what state biofilms were when they were inoculated in vivo. Were they sonicated prior to inoculation so only a few adherent colonies were present? Maybe clarify the steps indicating in what state biofilms were on microbeads for inoculation.

--Figure 1: What is the porous substrate on which the microbeads are sitting? The biofilm appears to be attached to it more so than the beads.

--p values needed throughout, for example lines 298 – 299 are needed. They refer to Figure 4, but it doesn’t have p values. The intensity signals look highly different, but authors suggest there is no significant difference (line 298).

--Figure 5: Are these six separate panels? Only label four of them with A, B, C, D.

--Overall, nicely done and informative.

6. PLOS authors have the option to publish the peer review history of their article (what does this mean?). If published, this will include your full peer review and any attached files.

Reviewer #1: No

---

## [Author Response · Author response to Decision Letter 0]

26 Oct 2020

Reply: We thank the academic editor and the reviewer for their careful reading of the manuscript and their constructive remarks. We have taken the comments on board to improve and clarify the manuscript. Please find below a detailed point-by-point response to all comments (line and page numbering refers to the revised manuscript).

Academic Editor: Please ensure that your manuscript meets PLOS ONE's style requirements, including those for file naming. 

Reply: The authors have made changes to the manuscript in order to agree with the PLOS ONE’s style requirements including those for file naming. Firstly, by adding an asterisk for the corresponding author (line 5). Then, by changing the order of department and institution (lines 6 and 8). The authors have changed the presentation of the corresponding author (lines 13 and 14) with the author’s initials. Postal codes have been removed (line 10). 

Concerning supporting information citation, we have changed lines 328 and 329 to conform to file naming guidelines. We have also changed the reference number for S1, S2, S3 movies in lines 567, 571, 575 respectively, and also S1, S2, S3, S4, S5, S6 Tables in lines 579, 582, 585, 588, 591, 594 respectively. According to these changes, corresponding files have been uploaded with their names changed.

Academic Editor: Please amend either the title on the online submission form (via Edit Submission) or the title in the manuscript so that they are identical.

Reply: The authors apologize for the difference between title on the online submission form and the manuscript. The term in vivo was missing in the title on the online submission form and it has been fixed with the PLOS ONE’s style requirements including those for file naming.

Comments to the Author:

1. Is the manuscript technically sound, and do the data support the conclusions?

Reviewer #1: Partly

Reply: In the responses to the reviewer’s comments, the authors have brought precise answers on technical aspects of their work and on statistical analyses performed, so that the raised conclusions were adapted to the data presented. The corresponding changes have also been made in the revised version of the manuscript.

2. Has the statistical analysis been performed appropriately and rigorously?

Reviewer #1: No

Reply : As described in the “Statistical analysis” section of the Materials and Methods, “Data generated were analysed with Prism 5 software (GraphPad Software, Inc.) and a non-parametric Mann-Whitney two-tailed statistical test. p≤0.05 was considered statistically significant (symbols: ****p≤0.0001; ***p≤0.001; **p≤0.01; *≤0.05; ns = non-significant)”. For experiments that required statistical analysis, the sentence “p≤0.05 was considered statistically significant (symbols: ****p≤0.0001; ***p≤0.001; **p≤0.01; *≤0.05; ns = non-significant)” was added to figure legends (figures 4 and 6).

5. Review Comments to the Author

Reviewer (Page 3 Line 64): Confirm context of the 80%. I believe it’s in developing countries, not in overall infections.

Reply: The authors agree with the reviewer’s remark concerning the lack of accuracy of the 80%. To go further, The National Institutes of Health (NIH) report that in developed countries, biofilms account for over 80 % of microbial infection in humans according to the report in 2002 entitled Research on Microbial Biofilms (Report No. PA-03-047). The authors have taken into account the lack of precision and have added “in developed countries” in lines 63 – 64. 

Reviewer (Line 138): What is 4.106? Is it 4 x 106, or 4.16? Just clarifying.

Reply: The authors have taken into account the mistake made in the number formatting. We have subsequently changed “4.106” to “4x106” in line 137 and also “1.35.108” to “1.35x108” in line 132.

Reviewer (Lines 137 – 145): What’s the rationale for disrupting the biofilms? If they grew on the surface, why not inoculate in their state of formation, rather than collection, spinning, and resuspending? By disrupting, the basal dormant layer it could make the shift to planktonic-type metabolism and not fully reflect a biofilm state.

Reply: The protocol used to recover biofilms after the 24-hour growth period consisted of a soft washing method involving steam, previously developed by Tasse et al in 2018. This method allowed us to preserve biofilm integrity and to remove planktonic bacteria not attached to the bottom of the well in the microplate. The biofilm was then recovered in a tube by flushing and scraping the bottom of the well. The resulting suspension was then washed twice by soft centrifugation at 100 x g at room temperature (changes made to line 142-143) in order to eliminate any biofilm not grown on microbeads. Then, the pellet was further resuspended in 50 µL of PBS to concentrate our inocula, as we microinjected a very small volume into the mouse ear pinna. Centrifugation was carried out at a low speed to limit any disruption of the biofilm ultrastructure and to avoid the shift to planktonic–type metabolism. The resulting suspension was defined as “biofilm-coated microbeads” in the manuscript and used as our inocula. 

Reviewer (Line 156): Were these biofilms imaged following the same resuspension, spinning, etc. process?

Reply: In this set of experiments, the authors wanted to observe the final inocula with as little modification as possible to the initial protocol described in the “Inoculum preparation” section in the Materials and Methods. More precisely, for observations of biofilm glucidic matrix components, an additional step of incubation with WGA-Alexa 488 and then washing was added before flushing and scraping the bottom of the well. The labelled biofilm was then washed twice by soft centrifugation, as described above. Ten µL of the final suspension was spread on a glass slide to be observed on the LEICA MM AF to evaluate microbeads coating efficiency.

Reviewer (Line 166): Same question; all processed the same prior to imaging?

Reply: In Figure 1, the authors wanted to verify that the “biofilm-coated microbeads” inoculum was not altered after passing through the 34G needle used for micro-injection (figures 1C-F), and if the sonication protocol used to disrupt biofilms was effective (figures 1G-H). Samples observed in figures 1C-D show biofilm-coated microbeads prepared as described above with the following sequential steps: steam washing, flushing, scraping and resuspension after soft centrifugation. A small volume (5 µL) was withdrawn with a pipette and deposited on SEM Pore filters (DTM9305, Jeol) corresponding to the condition “biofilm-coated microbeads before micro-injection”. After passive absorption on the filters, the samples were then treated as described in the Materials and Methods section (lines 169-177). Sample observations were made directly on filters. Figures 1E-F show samples that were prepared and treated exactly as described previously, except that samples were withdrawn and deposited on SEM Pore filters after slowly passing through a 34G needle fitted to a NanoFil syringe (World Precision Instruments) (corresponding to “biofilm-coated microbeads after micro-injection”). The images shown in figures 1C-F illustrate that the transit of the “biofilm-coated microbeads” inoculum in the needle of the syringe did not disrupt the biofilm ultrastructure.

In figures 1G-H, we wanted to show that our sonication protocol was able to dissociate all bacteria from the surface of “biofilm-coated microbeads”, in order to validate our biofilm titration protocol. Samples observed were prepared as described above (steam washing, flushing, scraping, resuspension after soft centrifugation), then sonicated for 10 minutes and vortexed for 30 seconds before and after this process. A small volume (5 µL) was withdrawn with a pipette and deposited on SEM Pore filters. After passive absorption on the filters, the samples were then treated as described in the Materials and Methods section (lines 169-177).

Reviewer (Line 186): Did the vortex process disrupt the biofilm? Do you think it was still in the same state as when growth completed?

Reply: In vivo labelling with the CDy11 probe was carried out on “biofilm-coated microbeads” samples prepared as described previously. After resuspension of biofilms in 50 µL of PBS, 10 µL of the diluted probe was added to the tube. The vortex was then set to minimal speed and the tube was vortexed for 2 seconds (gentle vortex) in order to minimize any alteration to the biofilm-coated microbeads. Images of CDy11 labelled biofilm-coated microbeads in the ear tissue (figures 3B and C) show that matrix components such as amyloid fibrils are still present in the inoculum, proving that the biofilm structure was not disrupted following the short vortex. These matrix components were less abundant for the planktonic form of the LYO-S2 strain of S. aureus, as previously shown in figures 1C-D for the LYO-S2 strain of S. aureus in Abdul Hamid et al 2020.

Reviewer (Line 228): How does steaming prevent biofilm detachment? This may be in the literature; I’m not familiar with it.

Reply: The technique used to wash biofilm-coated microbeads in our manuscript is a new steam-based method to investigate biofilms developed by Tasse et al in 2018. In their article, they presented a simple and robust method, adapted for microplates, in which steam is used as a soft washing method to preserve biofilm integrity and to improve reproducibility of biofilm preparation. Presented below is an illustration of the system from the paper. Biofilm-coated microbeads formed at the bottom of the well stay attached even after steam washing. We wanted to study innate immune responses elicited specifically against biofilms coated on microbeads and this method allowed us to eliminate any non-attached bacteria that have not made the transition towards biofilm-type metabolism. Thus, the aim of this steaming technique is to clear the inoculum from planktonic bacteria and not to prevent biofilm detachment.

Reviewer (Lines 253 – 257): This paragraph is confusing. Did you want the biofilms on or off the beads? As questioned above, it’s not fully clear in what state biofilms were when they were inoculated in vivo. Were they sonicated prior to inoculation so only a few adherent colonies were present? Maybe clarify the steps indicating in what state biofilms were on microbeads for inoculation.

Reply: The aims of figures 1 and 2 respectively were to qualitatively and quantitatively characterize the biofilm-coated microbeads. In figures 1A-B, we show images of uncoated microbeads compared to images of biofilm-coated microbeads (figures 1C-F). To ensure that the immune response elicited from both types of inocula (planktonic bacteria & microbeads or biofilm-coated microbeads) were comparable, we quantified the number of microinjected bacteria for each experiment. To do this, after inoculum preparation and micro-injection into the mouse ear pinna, we diluted an aliquot of the sample tenfold in PBS. The diluted biofilm-coated microbeads sample was then vortexed for 30 seconds followed by a 10-minute sonication and another vortex for 30 seconds. Inocula titrations were then determined by serial dilutions and plating on LB agar medium. CFUs were enumerated after 24 hours of incubation at 37°C. Since directly plating biofilm-coated microbeads on agar plates could induce bias during titration, we added a sonication step to our protocol and verified whether it was successful at detaching bacteria from the microbeads’ surface. Figures 1G-H show that bacteria were fully detached from microbeads after the sonication protocol.

In figure 2, we wanted to further characterize our different inocula. We first labelled sialic acid and N-acetylglucosaminyl residues that are found in the biofilm extracellular matrix by using WGA-Alexa Fluor 488 dye. We also performed an analysis to quantify the number of microbeads covered with biofilm using flow cytometry (FACS). Figure 2 shows the gates used to determine the proportion of uncoated microbeads (B), planktonic bacteria & microbeads (C) and biofilm-coated microbeads (D) populations. The inset in the top right corner of the left panels in figures 2B- D show gates used to select microbeads (appear in light blue). The right panels in figures 2B-D represent the subsequent counts for each population according to their fluorescence intensities. Thanks to this technique, we confirmed that we have a high percentage (82.5%) of biofilm–coated microbeads in our inocula as compared to the “planktonic bacteria & microbeads” sample.

The authors apologize for the confusion and have modified the manuscript from lines 247 to 260 to make it easier for readers to understand. 

Reviewer (Figure 1): What is the porous substrate on which the microbeads are sitting? The biofilm appears to be attached to it more so than the beads.

Reply: In SEM micrographs, the porous substrate in figure 1 on which microbeads are sitting correspond to the SEM PORE filters (DTM9305, Jeol) which have 500nm diameter holes (a picture of the filters is seen below). Biofilm microdroplets were deposited on these filters with a needle or a pipette and passively diffused through the filter. The biofilm was not grown on these filters, it was only deposited on them and then fixed for SEM analysis. Bacteria and beads are therefore not adherent to these filters. All samples shown were prepared as described in MM and then deposited on filters, which is essential for SEM observations.

The authors have revised the section “Observation of bacterial inocula by fluorescence microscopy, flow cytometry and scanning electron microscopy (SEM)” of the Materials and Methods (lines 169-177) to make it easier for readers to understand.

Reviewer: --p values needed throughout, for example lines 298 – 299 are needed. They refer to Figure 4, but it doesn’t have p values. The intensity signals look highly different, but authors suggest there is no significant difference (line 298).

Reply : The authors have added the p values for statistical analysis carried out in figure 4 (Lines 321 – 322) and 6 (Lines 374 – 376).

The protocol used to study the global inflammation in the inoculated ear tissue (figure 4) consisted of drawing a region of interest (ROI) around the EGFP+ zone of the late time-point image of a single experiment. The ROI was then saved and applied on the early time-point image of the same experiment. The sum of EGFP fluorescence intensities of each pixel in the ROI was measured for both time points and then divided by the area of the ROI. This step allowed us to normalize the fluorescence intensities between experiments. Each experiment was performed four times for each group of mice, which allowed us to create figure 4D. The images shown in figures 4A-C represent one of the four experiments performed for each group. Since the sum of fluorescence intensities was divided by the area of the ROI, this means that there is no direct correlation between the values presented in figure 4D and the fluorescence intensities of the images. Using this semi-quantitative method, we therefore measured the overall inflammation of the mouse ear pinna tissue over time. We have attached the raw data for each experiment as a supplementary table in the latest submission of our manuscript (S6 Table).

--Figure 5: Are these six separate panels? Only label four of them with A, B, C, D.

Reply: The aim of figure 5 is to analyze the immune response at the injection site by observing the cellular behavior against planktonic bacteria & microbeads or biofilm-coated microbeads. Specifically, we analyzed the behavior of recruited cells interacting with bacteria or not, at the injection site. This original in vivo study performed by intravital imaging allowed us to follow cell movement during a long period of time (30 minutes), at each time point and for each bacterial form. 

Figure 5 illustrates live confocal imaging after inoculation of planktonic bacteria & microbeads (A) or biofilm-coated microbeads (B) in the ear pinna of LysM-EGFP transgenic mice at early time points (4-6 hpi). Images show maximal intensity projections of EGFP (green) and mCherry (magenta) fluorescence at different time during the acquisition at T0 for each time point and 30 minutes later. 

Firstly, in figure 5A, innate immune cell recruitment after inoculation of planktonic bacteria & microbeads was observed between 4.35 and to 5.05 hpi. Secondly, in figure 5B, innate immune cell recruitment after inoculation of biofilm-coated microbeads was observed between 4.15 and 4.45 hpi. Live confocal imaging of two additional experiments at early time points after inoculation of biofilm-coated microbeads were provided in figures 5 C and D at 4.15 hpi and 4.00 hpi.

The authors apologize for the confusing annotation of confocal images and have revised the legend of figure 5 and have added a letter corresponding to each image on the figure to make it easier for readers to understand (Lines 336 - 346).

---

## [Decision Letter · Decision Letter 1]

23 Nov 2020

Biofilm-coated microbeads and the mouse ear skin: an innovative model for analysing anti-biofilm immune response * in vivo *

PONE-D-20-26002R1

Dear Dr. Gueirard,

We’re pleased to inform you that your manuscript has been judged scientifically suitable for publication and will be formally accepted for publication once it meets all outstanding technical requirements.

Kind regards,

Amitava Mukherjee, ME, Ph.D.

Academic Editor

PLOS ONE

Additional Editor Comments (optional):

Reviewers' comments:

Reviewer's Responses to Questions

**Comments to the Author**

1. If the authors have adequately addressed your comments raised in a previous round of review and you feel that this manuscript is now acceptable for publication, you may indicate that here to bypass the “Comments to the Author” section, enter your conflict of interest statement in the “Confidential to Editor” section, and submit your "Accept" recommendation.

Reviewer #1: All comments have been addressed

2. Is the manuscript technically sound, and do the data support the conclusions?

Reviewer #1: Yes

3. Has the statistical analysis been performed appropriately and rigorously? 

Reviewer #1: Yes

4. Have the authors made all data underlying the findings in their manuscript fully available?

Reviewer #1: Yes

5. Is the manuscript presented in an intelligible fashion and written in standard English?

Reviewer #1: Yes

6. Review Comments to the Author

Reviewer #1: (No Response)

7. PLOS authors have the option to publish the peer review history of their article (what does this mean?). If published, this will include your full peer review and any attached files.

Reviewer #1: No

---

## [Editor Report · Acceptance letter]

26 Nov 2020

PONE-D-20-26002R1 

Biofilm-coated microbeads and the mouse ear skin: an innovative model for analysing anti-biofilm immune response *in vivo*

Dear Dr. Gueirard:

I'm pleased to inform you that your manuscript has been deemed suitable for publication in PLOS ONE. Congratulations! Your manuscript is now with our production department. 

Kind regards, 

on behalf of

Professor Dr. Amitava Mukherjee 

Academic Editor

PLOS ONE